# Ex Vivo Expansion of Hematopoietic Stem Cells for Therapeutic Purposes: Lessons from Development and the Niche

**DOI:** 10.3390/cells8020169

**Published:** 2019-02-18

**Authors:** Parisa Tajer, Karin Pike-Overzet, Sagrario Arias, Menzo Havenga, Frank J.T. Staal

**Affiliations:** 1Department of Immunohematology and Blood Transfusion, L3-Q Leiden University Medical Center, 2333 ZA Leiden, The Netherlands; S.P.Tajer@lumc.nl (P.T.); K.Pike-Overzet@lumc.nl (K.P.-O.); 2Department of Molecular Cell Biology, Leiden University Medical Center, 2333 ZA Leiden, The Netherlands; 3Batavia Biosciences, Zernikedreef 16, 2333 CL Leiden, The Netherlands; s.ariasrivas@bataviabiosciences.com (S.A.); m.havenga@bataviabiosciences.com (M.H.)

**Keywords:** hematopoietic stem cell, ex vivo expansion, gene therapy, clinics, transplantation

## Abstract

Expansion of hematopoietic stem cells (HSCs) for therapeutic purposes has been a “holy grail” in the field for many years. Ex vivo expansion of HSCs can help to overcome material shortage for transplantation purposes and genetic modification protocols. In this review, we summarize improved understanding in blood development, the effect of niche and conservative signaling pathways on HSCs in mice and humans, and also advances in ex vivo culturing protocols of human HSCs with cytokines or small molecule compounds. Different expansion protocols have been tested in clinical trials. However, an optimal condition for ex vivo expansion of human HSCs still has not been found yet. Translating and implementing new findings from basic research (for instance by using genetic modification of human HSCs) into clinical protocols is crucial to improve ex vivo expansion and eventually boost stem cell gene therapy.

## 1. Introduction

HSCs comprise a small heterogeneous pool of cells initially formed during embryogenesis to maintain the blood system through a regulated process termed hematopoiesis along the lifetime of an organism [1,2]. HSCs are defined based on the unique dual capacity of self-renewal and multipotency, while the progenitors have restricted lineage differentiation and lack of self-renewal capacity. Hence, HSCs have become an attractive source for hematopoietic stem cell transplantations (HSCT) and regenerative medicine [3,4,5,6,7,8]. HSC quiescence, self-renewal and differentiation is controlled through extrinsic modulators largely provided by microenvironment, as well as by stem cell-intrinsic regulators [9]. One of the main limitations of HSC application for transplantations within the clinic is the limited quantities of HSCs collected from patients or donors [7,10,11]. A better understanding of stem cell biology and the mechanisms involved in HSC self-renewal in vivo is crucial for the development of ex vivo expansion protocols and subsequently for HSC-based gene therapy in clinical applications.

## 2. Hematopoietic Stem Cell Hierarchy

HSCs comprise a molecularly and functionally heterogeneous pool that gives rise to diverse blood and immune cells in a hierarchical manner. In the classical hierarchy model (Figure 1), multipotent HSCs are located at the top of the hierarchy and generate short-term HSCs or multipotent progenitors (MPPs), resulting in short-term multilineage repopulation [10,12,13,14,15]. The MPPs, at the same time, give rise to lineage-committed progenitors of common lymphoid (CLP) and common myeloid progenitors (CMP). Furthermore, CMP give rise to granulocyte/monocyte and Megakaryocyte/erythrocyte progenitors (MEP), which differentiate into platelets and red blood cells [16,17]. However, recent data from cell purification and functional assays in both human and mice challenge the current model and provide a new roadmap to describe the blood hierarchy [14,18,19,20]. These new insights based on single cell RNA sequencing analyses show common features between Megakaryocyte (Mk) and HSCs. Additionally, a study by Notta et al. demonstrated a shift in progenitor classes from embryo to adult. In this study, single cell functional analyses showed eminent granulocyte/monocyte, erythrocyte (Er) and Mk in fetal liver (FL); however, mainly Er and granulocyte/monocyte-committed progenitors were observed in bone marrow (BM). Moreover, they also showed Mk-Er-committed progenitors within the multipotent compartment, suggesting that Mk can differentiate directly from HSC, bypassing CMP [18]. Other studies, using limited dilution and single cell transplantation in mice, showed an HSC hierarchy model with different lymphoid and myeloid output [21,22]. The existence of a platelet-biased HSC was first identified in mouse model. It has been suggested that this population resides at the apex of the hierarchy, with a tendency for short- and long-term reconstitution of platelets in mice [14]. Also, Yomamoto et al. identified a subset within phenotypically defined HSCs that comprised functionally myeloid-restricted repopulation progenitors (MyRPs). Thus, they demonstrated that HSCs could give rise directly to MyPRs through a myeloid-bypass pathway (Figure 1) [12].

In addition, current advances in fluorescence-activated cell sorting (FACS) and sorting strategies provide high-purity isolation and identification of HSCs and progenitors using various cell surface markers. For instance, CD34, CD38, CD90, CD45RA and CD49f are common surface markers used for identifying human HSCs and progenitors in vitro and in vivo [7]. However, the expression of some of these markers such as CD38 of CD90 can change in vitro. Therefore, identifying robust stable markers that support the identification of HSCs subsets is crucial, especially when testing novel expansion protocols [23]. Novel surface markers have been suggested for identification of HSCs subsets; for instance, junction adhesion molecule-2 (Jam2) is highly expressed in a HSC subset that preferentially generates T cells [24]. Endothelial cell-selective adhesion molecule (ESAM) is another reliable marker for identification of both murine and human hematopoietic stem cells that are expressed throughout the lifetime. ESAM is highly expressed in long-term HSCs and MPPs. However, disruption in ESAM leads to an increased generation of T cells versus B cells. Therefore, ESAM may influence HSC differentiation paths [25,26,27,28]. Recently, endothelial protein C receptor (EPCR) was identified as a relatively robust surface marker for murine and human LT-HSCs [23].

Broad use of single cell RNA sequencing (scRNA-seq) has also helped in identifying different cell populations within HSC pools by screening thousands of gene expression profiles of single HSCs and progenitors [29,30]. This method can be used for different purposes, such as identifying new progenitor populations, cellular hierarchies in normal and distorted hematopoiesis or distinction between self-renewal potential and activation of lineage programs [31].

Dormant HSCs have been identified in mice using label-retaining studies and flow cytometry. Dormant HSCs divide four to five times throughout their lifetime, and they retain multilineage self-renewal capacity, while remaining quiescent during hematopoiesis. Label-retaining studies suggest that these cells can switch reversibly from dormancy to self-renewal in response to hematopoietic stress [13,32]. Therefore, heterogeneity in self-renewal capacity seems to be linked to HSC dormancy. Several label-retaining assays have been developed to isolate the HSCs based on their division history. However, one of the major limitations linked to this method is indirect measurement of symmetric divisions. Thus, distinct cell cycle properties within the HSC pool are intimately linked to HSC function.

Therefore, exploration of new and robust surface markers using FACS and sorting strategies, as well as single cell RNA sequencing, will provide a better understanding of both the molecular mechanism and intrinsic programming of HSCs, thus shedding light on their heterogeneity, lineage choice, functionality and the impact on ex vivo culturing.

## 3. HSC Self-Renewal Regulation and Niche

How HSC self-renewal is regulated remains one of the main questions of the hematopoietic stem cell biology field, and answering it will bring new knowledge and possibilities beneficial for clinical applications. Self-renewal, which is important in maintenance of HSC pool size, is regulated by key genes and proteins such as transcription factors, epigenetic modifiers and cell cycle regulators, as well as extrinsic factors from the environment, called niche [33,34,35,36]. Niche provides a specialized and tightly regulated environment that determines the stem cell fate, regulates the proliferation rate and protects cells from exhaustion and cell death [1,37,38,39]. Identification of stem cell niches, using advanced imaging techniques and genetic tools, has shed light on stem cell regulation and hematopoiesis in murine model (reviewed in [40]). However, little is known about the key cell types and growth factors involved in communication between human HSCs and the niche, due to challenges of modeling this network, as well as limited materials in the human system and obvious ethical constraints.

HSCs emerge at several sites at different developmental stages. During embryogenesis, definitive hematopoiesis in mice starts at the aorta-gonad-mesonephros region (AGM) and then moves to the fetal liver (FL) and eventually to bone marrow (BM), where the HSCs retain fetal liver characteristics for three weeks, and finally, they mainly remain in a quiescent state and maintain their pool size by the regulation of HSC-self renewal and differentiation. They are highly proliferative in FL and undergo several symmetrical divisions to give rise to the HSC pool required for the lifetime [10,11,41,42,43]. BM is a primary site of hematopoiesis in adult mammals after birth, although it can transfer partially into liver and spleen in response to severe hematopoietic stresses. It has been shown that HSCs in different BM regions have different self-renewal capacities [44]. Quiescent HSCs are mainly located near perivascular stromal cells and arterioles with low level of oxygen in BM [1,45,46] (Figure 2). Moreover, Kohler et al. showed the dynamics of young and aged HSCs inside the BM cavity, demonstrating that the location of HSCs in BM is changed during ageing [47]. Additionally, advanced imaging techniques, in combination with conditional depletion of important regulators in mouse models, have helped in identification of key cells and local signals and their role in HSC maintenance [48].

Besides this, the BM niche is heterogeneous, with a diversity of cell types in mice and humans. Endothelial cells, mesenchymal stromal cells (MSCs), megakaryocytes, osteoblasts and nerve cells within BM have been shown to influence self-renewal and expansion of murine and human HSCs (extensively reviewed in [49,50]), directly or indirectly.

A better understanding of HSCs interaction with their niche will help to overcome the high need for HSCs in clinical applications. Recently, a bio-mimic 3D model of HSC niche has been developed using hydrogel. Engineering niche in 3D is promising for culturing and studying the essential niche factors under steady state and active conditions of HSCs with better control of HSC behavior in vitro [51].

## 4. Wnt Signaling in HSC

The Wnt pathway is a prominent component in self-renewal of adult stem cells and fetal hematopoiesis. The Wnt signaling cascade has several different signal transduction possibilities, referred to as canonical (Wnt/β-catenin) and non-canonical pathways [52,53]. Both pathways are involved in complex processes, such as embryonic development, stem cell maintenance and tissue homeostasis. The well-understood canonical pathway is involved in cell fate, proliferation and survival, while the non-canonical pathway is more involved in differentiation, apoptosis regulation, cell polarity and migration [53]. β-catenin is a key player in the canonical pathway. In the absence of Wnt ligands, the level of β-catenin remains very low in cytoplasm through the action of a protein complex called the destruction complex, which actively degrades β-catenin. At the cell membrane, the interaction of Wnt ligands and Frizzled receptors and co-receptors leads to inactivation of destruction complex and subsequently to high level of β-catenin and its translocation to the nucleus. In another way, non-canonical pathways can be activated through Ca^2+^ signals, JNK kinases or other receptors such as Ryk, besides Wnt/Frizzled interaction [53,54,55]. Given the importance of Wnt signaling in hematopoiesis, there is great interest in exploiting this pathway for ex vivo expansion of HSCs [56]. The potential role for Wnt signaling in the self-renewal of HSC and proliferation of progenitors has recently been extensively reviewed [57].

Ex vivo culturing of HSCs using non-canonical wnt5A protein showed increased HSC repopulation in mice [58]. Also, introducing wnt3a proteins increased murine HSC self-renewal in vitro [59]. Constitutive expression of β-catenin enhanced the HSC self-renewal in mice [60]. Moreover, given that prostaglandin E2 (PGE2) affects β-catenin stability, Zon et al. suggested using PGE2, which induces canonical Wnt signaling [61] for ex vivo modulation of human cord blood HSC [62]. Also, activation of Wnt/ β-catenin pathway using glycogen synthase kinase 3 (GSK3-β) inhibitor in combination with Rapamycin inhibitor of mTOR pathway could increase the number of murine LT-HSCs in vivo [63]. These studies suggest that similar strategies could be taken for ex vivo maintenance of human and murine LT-HSC.

The role of wnt/ β-catenin signaling in HSC self-renewal is complex and is dependent on factors initiated from niche [64]. The function of Wnt signaling is strictly controlled in a dosage-dependent fashion. High levels of Wnt in HSCs push stem cells into exhaustion and limited reconstitution in irradiated patients by driving HSC differentiation toward mature blood lineage and loss of proliferation, while lower doses of Wnt result in better maintenance of immature cells and higher long-term repopulation capacity.

## 5. Notch Signaling in HSC

Notch signaling is another crucial conserved pathway in cell fate decisions, development and hematopoiesis [65]. The role of Notch in the regulation of HSCs in mammalian is still controversial. In mice, Notch signaling is essential for early HSC development at AGM during embryogenesis, while its activity is reduced during HSC maturation, and it becomes unnecessary for adult HSC maintenance in BM [66]. In mice, Notch signaling plays an important role in the development of early HSCs by establishing the vascular and circulating system, which are essential for HSC development. Notch signaling also promotes the expression of Notch receptors and ligands to drive HSC development [67,68]. Notch signaling is based on cell–cell interactions, where the Notch receptors interact with the transmembrane ligands of Delta and Jagged. This interaction leads to proteolysis of the receptor and release of the Notch intracellular domain. The released domain is then translocated to the nucleus, where it forms a protein complex with transcription factors and binds the regulatory elements of Notch target genes and activates their expression [65,69].

Early studies on Notch signaling in mice using the gain-of-function method also suggested its role in HSC self-renewal. Overexpression of Notch genes or receptors in HSCs increased the number of functional HSCs, self-renewal capacity of HSCs, and inhibited differentiation [70,71,72]. On the other hand, loss of function experiments, such as conditional knock out of Notch receptors (Notch 1, 2) or ablation of ligands and regulators, did not show any effect on HSC self-renewal [71,73]. Souilhol et al. showed that both Notch 1 and Notch 2 are involved in early development of HSCs, even though the Notch 2 signal is weaker than Notch 1, suggesting that modulation of Notch signaling can be considered for HSC generation from endothelial stem cells or induced pluripotent stem cell for clinical applications. Moreover, recent studies on human BM have shown that niche signals can activate Notch signaling in HSCs and progenitors. However, other studies showed embryonic lethality and impaired hematopoiesis in mice with targeted mutations in Notch 1 or Jagged 1. No impairment of hematopoiesis was observed in Notch 2, 3, 4 knock outs, suggesting their non-essential role in hematopoiesis [66]. Therefore, while a functional role for Notch signaling in embryonic development has been established, the need for Notch signaling in adult hematopoiesis, certainly under hemostasis, is more controversial [74]. 

In evolution, these conserved pathways crosstalk to ensure a correct and strong control of gene expression. In this manner, Notch and Wnt pathways crosstalk on cell fate decisions. Studies in skin and mammary glands indicated that Wnt signaling controls the stem cell maintenance, whereas Notch promotes lineage commitment and differentiation [75,76,77,78]. 

## 6. Ex Vivo Expansion of HSC

Allogeneic HSC transplantation (HSCT) has been used efficiently in the clinic for patients with severe immunodeficiency diseases with a matched donor. However, for patients without a suitable donor, genetically modified autologous HSCs have been applied with low risk of graft-versus-host disease, which is the main cause of transplant morbidity and mortality [79,80,81,82]. One of the main challenges of using HSCs in clinical application is the limited number of cells that can be enriched from the patient. Thus, expanding HSCs has become important, due to the increasing interest in stem cells and the gene therapy field, particularly for clinical applications. HSCs undergo symmetrical and asymmetrical cell divisions in vivo. The frequency of symmetric and asymmetric cell divisions determines the number of stem cells and differentiated cells in niche. Symmetric cell division leads to the expansion of HSCs, in numerical terms. Thus, for ex vivo expansion, approaches that will result in symmetric stem cell division and self-renewal without further differentiation are required [83]. Different combinations of recombinant growth factors and cytokines have been assessed to expand HSCs ex vivo [84,85], however, limited success in clinical studies has been reported due to unwanted differentiation of stem cells and progenitor inputs. Currently, combinations of different cytokines and growth factors, such as SCF, Flt3, TPO, IL-3, and IL-6, are commonly used for supporting HSC and progenitor survival, proliferation and maintenance during in vitro culturing systems in clinics (reviewed in [86,87]). The influence of cytokines on lineage commitment and self-renewal has been studied extensively. A recent study by Knapp et al. indicates that this cocktail may only regulate short-term (4 days) survival and proliferation of human HSCs, rather than maintenance of functional long-term HSC in vitro [88]. Unsuccessful attempts to improve the HSC engraftments using current cytokine-based ex vivo expansion protocols clearly suggest the need for additional factors to support in vitro HSC maintenance and expansion.

A high-throughput screening of thousands of molecules has been used to find new compounds by testing their potential for in vitro stem cell expansion. Among those molecules, Prostaglandin E2 (PGE2), Stemregenin 1 (SR1) (an Aryl hydrocarbon receptor antagonist) and UM171 were found through library screenings on CD34^+^ cells and further tested due to their potential in expanding HSCs in vitro [89,90,91].

PGE2 has been investigated for optimizing in vitro expansion of human HSCs for clinical applications. The outcome of clinical trials using optimized and shortened ex vivo expansion of HSCs with PGE2 resulted in no effect on ex vivo proliferation and colony forming potential. However, enhanced and rapid engraftment in human cord blood transplantation, as well as enhanced lentivirus transduction efficiency, was reported in this study. PGE2 can be considered for clinical hematopoietic stem cell-based gene therapy, as these findings suggest that short-term modulation of HSCs with PGE2 can overcome some challenges observed during ex vivo expansion of HSCs, such as unwanted differentiation due to long-term expansion of HSCs and high manufacturing costs involved in clinical trials [62,92,93,94].

SR1 was first identified for its ability to support expansion of both murine and human CD34^+^ in vitro, with faster recovery of neutrophil and platelet in vivo [89]. Therefore, the clinical potential of SR1 has been explored by culturing CD34^+^ in presence of TPO, SCF, Flt3 and IL-6 [90]. However, recent clinical and phenotypic/transcriptional studies demonstrated the increase of multipotent progenitors and erythroid/megakaryocytic in cultured CD34^+^ with SR1, rather than long-term repopulating stem cells [90,95].

UM171 is a promising candidate for ex vivo expansion of HSC for allogenic transplantation and gene therapy, and a clinical trial using UM171 for allogenic stem cell transplantation is currently ongoing (NCT02668315). This molecule has been shown to improve ex vivo expansion of human cord blood HSCs with long-term repopulation potentials [91], while also increasing the myeloid progenitors in ex vivo culture [95]. However, it suppresses megakaryocyte/erythrocyte and granulocyte expansion, although this suppression effect could be counteracted by adding SR1. In addition, a recent study has reported the ability of UM171 to enhance gene transfer to HSCs in a dose-dependent manner by up to 2-fold, with increased recovery of transduced HSC in mice [96]. All of this together suggests the potential of UM171 and SR1 for ex vivo gene therapy applications.

It is already known that epigenetic regulation such as DNA methylation and post-translational histone modifications play an important role in the cell-fate decisions of HSCs. These modifications allow each cell type to acquire unique forms and functions. Distinct epigenetic markers control gene expression [97,98,99,100]. For instance, during development, the DNA methylation patterns are not conserved. Changes in H3K27me3 and the enhancer H3K27ac are among the well-known epigenetic changes through embryonic stem cell development [101,102,103]. H3K4me3 is another epigenetic modification in promoters associated with gene transcriptional activation. Loss of H3K4me3 leads to reduced self-renewal and impaired differentiation in murine embryonic stem cells [104,105]. The role of H3K4me3 in gene expression is not completely clear; however, it is known that H3K4me3 is involved in the transcriptional machinery assembly and facilitates induction of gene expression in response to the microenvironment [106,107]. Given the importance of epigenetic regulation in HSC regulation, multiple studies have demonstrated that small molecule inhibitors of histone deacetylase (HDAC) and DNA methyltransferase have the capability to support ex vivo expansion of HSCs. Culturing human cord blood-derived CD34^+^ with valproic acid, the HDAC inhibitor showed the upregulation of genes involved in stemness, and increased the SCID repopulating cells [108]. However, some methyltransferases, such as G9a and G9a-like protein (GLP), support lineage choice and differentiation. Introducing small molecule inhibitors of methyltransferases and HDACSs can have a synergetic effect on ex vivo expansion, helping the maintenance—but not the expansion—of HSCs in culture. For example, combination of the decitabine—a methyltransferase inhibitor—and trichostain A—a HDAC inhibitor—showed greater ability to maintain HSC activity ex vivo than individual single agents [109,110,111].

The discovery of several new small molecules in recent years has taken HSC expansion beyond traditional hematopoietic cytokine treatment, and these compounds are currently making their way into clinical trials. These approaches are summarized in Table 1.

## 7. Concluding Remarks

The number of HSCs that can be obtained for clinical transplantation is limited and influenced by different factors, such as a low cell number of HSCs isolated from patients. Thus, expansion of HSCs for gene therapy and regenerative medicine is an unmet medical need. Optimizing current cell culture methods for clinical applications by using, for instance, proteins from conserved pathways such as Wnt and Notch ligands, or small molecules such as PGE2, SR-1 or UM171, might improve the ex vivo expansion of HSCs and also increase the engraftment capacity of gene-modified HSCs [91,93,115].

Enrichment of HSCs for clinical applications is mainly based on CD34^+^ selection. However, the CD34^+^ population is heterogeneous, with a small fraction of LT-HSCs that are crucial in clinical therapy. Hence, enrichment using only CD34^+^ is not sufficient. Advances in FACS, sorting strategies and single cell RNA sequencing have allowed identification and isolation of different high-purity subpopulations within the HSC pool. Further enrichment of HSCs using well-known established markers such as CD90^+^, CD45RA^+^ and CD49f^+^ would help to target desired populations, avoiding unwanted material loss. Even though the surface phenotype of cultured HSCs differs from that of unmanipulated HSCs, identifying and including robust surface markers such as EPCR could improve in vitro measurements of HSCs [23].

Enormous efforts have already been invested in identifying the key cellular players for HSC self-renewal regulation in the niche. Imaging advances and genetic tools have increased the general knowledge of the HSC niche. Clarification of the interactions between HSCs and their microenvironments may help to identify novel clinical approaches and opportunities in the HSCT field. Therefore, a better understanding of the mechanisms involved in HSC regulation and fate determination could also help in the development of new strategies for ex vivo expansion. Given that gene repair approaches (using for instance CRISPR/Cas9), as opposed to than gene addition approaches, require more extensive ex vivo culturing of HSCs, developing such strategies is important for clinical implementation of new gene therapy methods.

## Figures and Tables

**Figure 1 cells-08-00169-f001:**
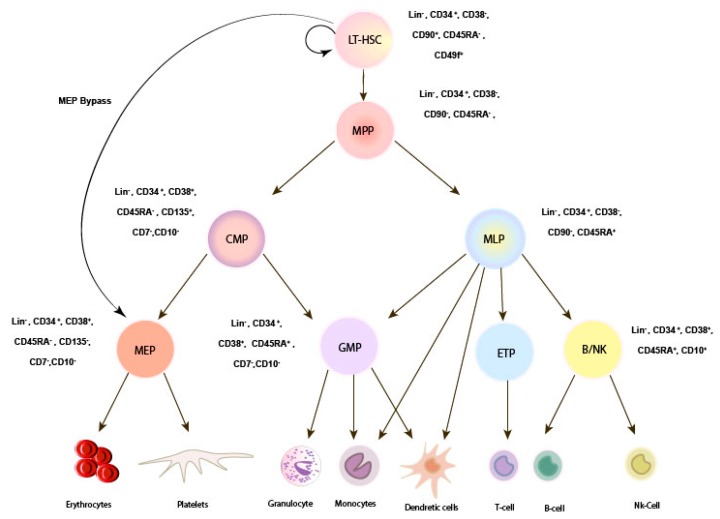
Revised model for human HSC hierarchy. In the classic model for the human HSC hierarchy LT-HSCs are defined by CD34+ CD38- CD45RA- CD90+CD49f+ which differentiates into MPPS, CMPs, MLPs, GMPs. However, in a revised model, HSCs can differentiate directly into MEPs by bypassing CMP (here shown as MEP bypass route). LT-HSC: long-term hematopoietic stem cell. MLP: multipotent progenitor, CMP: common myeloid progenitor, GMP: granulocyte/macrophage progenitor, MEP: Megakaryocyte-erythrocyte progenitors.

**Figure 2 cells-08-00169-f002:**
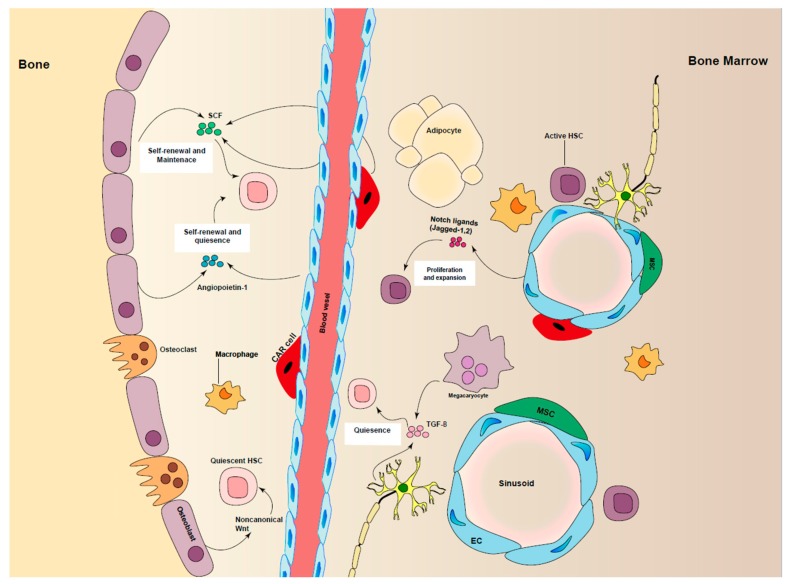
Regulation of HSCs maintenance in niche. Different cell types are involved in promoting HSC maintenance, including perivascular stromal cells, endothelial cells (ECs), macrophages, CAR cells, sympathetic neurons by producing cytokines and growth factors such as stem cell factor (SCF), angiopoietin-1, TGF-β and notch ligands.

**Table 1 cells-08-00169-t001:** Summary of current protocols of ex vivo expansion of human HSCs.

Factor	Components	Supplements	Input Cells	Culture Time	Effects	References
Cytokine supplement	-	SCF, FLt3, TPO, IL3	CD34+	7 days	20-fold expansion of CD34+ in vitro Similar frequency of human CD45+ BM cells vs. fresh cells (NOD/SCID)	[112]
	-	SCF, FLt3, TPO, IL3, IL-6	CD34+CD38-	4 days	15-fold increase in CFUs, and fourfold enhanced chimera	[113,114]
	-	SCF, TPO, FGF-1, IGFBP-2, ANGPTL5	CD133+	11 days	230-fold increase in TNCs in vitro	[114]
	Notch ligands		CD34+	14–21 days	Neutrophil recovery and myeloid engraftment	[56]
Chemical supplement	PGE2	-	CD34+	24–48 h	Enhances neutrophil recovery enhancing homing, survival, and proliferation of HSCs	[93]
	SR1	SCF. Flt3L, TPO, IL-6	CD34+	7–21 days	65-fold increase in CFUs; 17-fold enhanced chimera; Enhances neutrophil recovery	[89,90]
	UM171	SCF. Flt3L, TPO	CD34+	7–21 days	More than 100-fold expansion of LT-HSC, and 35-fold enhanced chimera; Inhibiting erythroid and megakaryocytic differentiation	[91]
	Histone deactylase inhibitor (valproic acid)	SCF. Flt3L, TPO, IL-3	CD34+	7 days	36-fold increase in SCID-repopulating cells; improving homing and maintaining quiescence	[108]
	DNA Methyltransferase inhibitor (UNC0638)	SCF. Flt3L, TPO, IL-6	CD34+	2 weeks	Maintaining HSC activity blocking formation of higher-order chromatin structure	[109]

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
