# Peer review of "Ex Vivo Expansion of Hematopoietic Stem Cells for Therapeutic Purposes: Lessons from Development and the Niche"

_cells, 2019, doi:10.3390/cells8020169_

Round 1

Reviewer 1 Report

This review paper is providing comprehensive information about the biology and clinical application of hematopoietic stem cells (HSCs) covering their surface markers, revised model of hematopoiesis, mechanisms regulating their self-renewal, their niche in the bone marrow, Wnt signaling, Notch signaling, epigenetic regulation, and recent clinical trials of HSCs artificially expanded ex vivo. This reviewer believes that this review will help physicians as well as researchers in the hematology field to catch up with very recent knowledge about HSCs. Only a couple of minor comments are listed below.

1.    Page 2 line69-72

Also, high expression of endothelial cell-selective adhesion molecule (ESAM) is normally found in murine and human HSCs committed to develop into the megakaryocyte and erythrocyte lineages rather than in myeloid and lymphoid lineages [25,26]. 

----- I am afraid that this information is not correct because, while ESAM seems to be functionally important in the megakaryocyte and erythrocyte lineages, high expression of this molecule marks authentic HSCs with robust differentiation potentials to the myeloid-lymphoid lineages as well as erythroid-megakaryocyte (Ooi et al Stem Cells 2009, Yokota et al Blood 2009). Please refer the 2 original papers that identified ESAM as the reliable HSC marker in addition to Ref#25-26 and amend the sentence in line 69-72.

2.    This reviewer has been aware of several misinformation in the references. (e.g.; References #99 and #110 are duplicating. #99 and #105 were not published in Cell.) Please go through the references carefully. 

Author Response

Dear editors,

We thank the reviewers for their useful and timely critique to our review article. Some reviers are of the opinion that our manuscript adds little to the literature. However, we here try to give a comprehensive overview of expansion strategies for human HSCs that are targets for gene therapy. This is becoming more relevant with gene repair efforts using nucleases such as CRISPR/Cas9, where HSCs need multiple rounds of division. We therefore think the review is timely and useful.

Reviewer 1:

This review paper is providing comprehensive information about the biology and clinical application of hematopoietic stem cells (HSCs) covering their surface markers, revised model of hematopoiesis, mechanisms regulating their self-renewal, their niche in the bone marrow, Wnt signaling, Notch signaling, epigenetic regulation, and recent clinical trials of HSCs artificially expanded ex vivo. This reviewer believes that this review will help physicians as well as researchers in the hematology field to catch up with very recent knowledge about HSCs. Only a couple of minor comments are listed below.

Page 2 line69-72

Also, high expression of endothelial cell-selective adhesion molecule (ESAM) is normally found in murine and human HSCs committed to develop into the megakaryocyte and erythrocyte lineages rather than in myeloid and lymphoid lineages [25,26]. 

----- I am afraid that this information is not correct because, while ESAM seems to be functionally important in the megakaryocyte and erythrocyte lineages, high expression of this molecule marks authentic HSCs with robust differentiation potentials to the myeloid-lymphoid lineages as well as erythroid-megakaryocyte (Ooi et al Stem Cells 2009, Yokota et al Blood 2009). Please refer the 2 original papers that identified ESAM as the reliable HSC marker in addition to Ref#25-26 and amend the sentence in line 69-72.

We thank the reviewer for this insight. The phrases have been adjusted as below and original references were added:

“Endothelial cell-selective adhesion molecule (ESAM) is another reliable marker for identification of both murine and human hematopoietic stem cells which is expressed through life time. ESAM is highly expressed on long-term HSCs and MPPs. However, disruption in ESAM leads to an increased generation of T cells versus B cells. Therefore, ESAM may influence the HSC differentiation paths”.

2.    This reviewer has been aware of several misinformation in the references. (e.g.; References #99 and #110 are duplicating. #99 and #105 were not published in Cell.) Please go through the references carefully. 

Thanks for your keen observation. Repeated references have been removed, and reference #99 and #105 are corrected to the correct journal.

Reviewer 2 Report

The manuscript "Ex vivo expansion of hematopoietic stem cells for therapeutic purposes: lessons from development and the niche" by Tajer and colleagues reviews the literature on methodologies to expand human hematopoietic stem cells for clinical application. As the authors state, this has been the holy grail of HSC research for several decades. Unfortunately, this topic has already been extensively discussed in a large number of previous reviews, often in more detail (Dahlberg et al., Blood, 2011; Flores-Guzman et al., 2013, Stem Cells Trans Med.; Xie and Zhang, Sci China Life Sci, 2015). In addition, it is not clear what are the novel insights or alternative views on existing controversies that this work brings, thus it does not seem to provide additional value to the already existing literature. 

Minor comments:

Line 79: message not clear

Lines 84-86: the take-home message of this sentence, as well as the link to previous sentence is not clear

Lines 132-133: this is not entirely true since up to 3 weeks of age, bone marrow HSCs retain fetal-liver characteristics (Bowie et al., PNAS, 2007).

Lines 162-171: text repeated in lines 135-143

Lines 196-197: What does the current review add on previous literature then?

Lines 244-245: Authors state published literature that leads to contradicting results, that it is not clear what the "importance of Notch signaling in the microenvironment for normal hematopoisis" is. Authors could try to provide an explanation. 

Lines 336-337: sentence is confusing

Line 360: Table 1 is not mentioned in the main text

Line 367: Spelling error on authors name

Line 393: seems inconsistent with other references

Author Response

Reviewer 2:

The manuscript "Ex vivo expansion of hematopoietic stem cells for therapeutic purposes: lessons from development and the niche" by Tajer and colleagues reviews the literature on methodologies to expand human hematopoietic stem cells for clinical application. As the authors state, this has been the holy grail of HSC research for several decades. Unfortunately, this topic has already been extensively discussed in a large number of previous reviews, often in more detail (Dahlberg et al., Blood, 2011; Flores-Guzman et al., 2013, Stem Cells Trans Med.; Xie and Zhang, Sci China Life Sci, 2015). In addition, it is not clear what are the novel insights or alternative views on existing controversies that this work brings, thus it does not seem to provide additional value to the already existing literature. 

The authors realize that the field is frequently reviewed by diverse research groups as HSC science is one of the hottest topics in life sciences today. As such there are constant new scientific findings which requires a constant flow of information and here review articles are an important means to keep the scientific community updated. Our review article addresses latest insights in human HSC culture and expansion in the context of gene therapy approaches and therefore the authors believe that publication of this manuscript is warranted as it will provide a valuable source to scientists and healthcare workers in the field.

Minor comments:

Line 79: message not clear

“Heterogeneity in self-renewal capacity seems to be linked to the HSCs dormancy”.

More explanation was added to the text, to clarify the message.

“Dormant HSCs have been identified in mice using label-retaining studies and flowcytometry. Dormant HSCs divide four to five times through life time and they retain multilineage self-renewal capacity, while they remain quiescent during hematopoiesis. Label retaining studies suggest that these cells can switch reversibly from dormancy to self-renewal in response to hematopoietic stress”.

Lines 84-86: the take-home message of this sentence, as well as the link to previous sentence is not clear

The text was rephrased to clarify the message as bellow:

Thus, exploration of new and robust surface markers using FACS and sorting strategies, as well as single cell RNA sequencing  will provide a better understanding of both the molecular mechanism, and intrinsic programming of HSCs thus, shedding light on their heterogeneity, lineage choice, functionality and the impact on ex vivo culturing.

Lines 132-133: this is not entirely true since up to 3 weeks of age, bone marrow HSCs retain fetal-liver characteristics (Bowie et al., PNAS, 2007).

The remark has been incorporated, with the caveat that is referring to mouse rather than human HSCs; the suggested reference has been included as well.

Lines 162-171: text repeated in lines 135-143

Redundant texts have been removed

Lines 196-197: What does the current review add on previous literature then?

In our current review we are try to give a comprehensive overview of expansion strategies for human HSCs that are target for gene therapy. This is also becoming more relevant with gene  This is becoming more relevant with gene repair efforts using nucleases such as CRISPR/Cas9, where HSCs need multiple rounds of division. application in gene therapy field. While the main focus of previous reviews was on the role of Wnt pathway in hematopoiesis and or T cell development.

Lines 244-245: Authors state published literature that leads to contradicting results, that it is not clear what the "importance of Notch signaling in the microenvironment for normal hematopoisis" is. Authors could try to provide an explanation. 

In the first paragraph, In line 237 we discussed the importance of NOTCH in early HSC development at AGM, also establishing vascular and circulating system which is needed for HSC development. However, the need for NOTCH signaling in adult hematopoiesis, certainly under hemostasis is more controversial. We refer to a recent review ( Lampreia FP, Carmelo JG, Anjos-Afonso F. Notch Signaling in the Regulation of Hematopoietic Stem Cell. Curr Stem Cell Rep. 2017;3(3):202-209)

Lines 336-337: sentence is confusing:

“Identification of several newly small molecules in recent years are going beyond traditional hematopoietic cytokines and making their way to the clinic”

The sentence is rephrased as below:

“The discovery of several new small molecules in recent years has taken  HSC expansion beyond traditional hematopoietic cytokines treatment and these compounds are currently making their way into clinical trials”

Line 360: Table 1 is not mentioned in the main text

The reference error is fixed

Line 367: Spelling error on authors name

Error has been corrected

Line 393: seems inconsistent with other references

MDPI format provided by journal has been used. 

Reviewer 3 Report

1. Given the number of reviews on HSC expansion, in what specific way(s) do the authors envision that this review will add to the existing literature?  For example, how does this review build on previous reviews such as by Bari et al (2015 - Expansion and Homing of Umbilical Cord Blood Hematopoietic Stem and Progenitor Cells for Clinical Transplantation)? Is there a specific gap that the authors would like to fill with this review?

2. The authors have repeated a number of sentences in sections 3, 4, and 5. For example: line 96-98 is the same as line 118-120; line 99-100 repeats in line 131; and line 122-126 repeats in 155-160. The sentences are identical or nearly identical. Given the length of the review, you should avoid repeating sentences like this. Consider combining the sections or elaborating on the original idea without “copy and paste”.

3. Potential typos: line 86 - do you mean “shedding” instead of “shading”? ; line 220 - do you mean “mammals” instead of “mammalian”?; line 242 - do you mean “knock outs” instead of “knocked outs”?; line 257 – do you mean murine “embryo” instead of “embryonic”; line 337 - Fix reference error.

4. Section 9, line 268 – Why do you use Ref 97 and 99 as examples of genetically modified HSC. These studies use small molecules. 

5. On page 2, line 206, you discuss PGE2.  In your discussion and reference to Zon’s work you imply that they used PGE2 in an expansion culture.  The benefit of their strategy is to avoid the need for ex vivo cell expansion. The motivation for a “brief” treatment was that it might not require GMP cell manufacturing facility, which would reduce costs.  Their more recent clinical trial paper is here: Blood 2013 :blood-2013-05-503177; doi: https://doi.org/10.1182/blood-2013-05-503177 . Please modify your discussion to highlight these different strategies.  You should also consider grouping PGE2 differently in your table, as it is not used precisely in same manner as SR-1 for example. This more recent publication also discusses the possible mechanisms by which brief PGE2 treatment of cells might be enhancing transplant outcomes.

6. Regarding Section 10, Concluding remarks: This section is difficult to understand. You may like to consider a different paragraph structure. First paragraph (conclusion) can summarise the unmet needs that would benefit from technologies that enabled in vitro HSC expansion (transplant and gene therapy applications).  Second paragraph – you could move your discussion about the development of better marker profiles to enrich for HSC, deplete specific cell subsets or to identify HSC in expansion products to this paragraph.  Third paragraph – in this paragraph you could consider giving your current optimistic summary of our expanded knowledge of niche biology and expansion strategies and how cumulatively this is likely to help us to identify new pathways forward.

7. With respect to your current concluding remarks, I have questions/comments:

First paragraph – It is not clear what you are proposing regarding the surface markers. Do you propose that the input “HSC population” is wrong or not sufficiently enriched? Or are you saying that we cannot validate the engrafting population after expansion? Is it not useful to expand the progenitor population, which has been shown to provide short-term myeloid support after transplantation? Is it possible that EPCR is specific to UM171 expansion - do you know if this has been validated in other systems? What aspect of this statement refers back to Ref 55? Is there any more recent work?

Second paragraph: 

Line 351 – Consider using the word “identify” instead of “bring up”.

Author Response

Reviewer 3:

Given the number of reviews on HSC expansion, in what specific way(s) do the authors envision that this review will add to the existing literature?  For example, how does this review build on previous reviews such as by Bari et al (2015 - Expansion and Homing of Umbilical Cord Blood Hematopoietic Stem and Progenitor Cells for Clinical Transplantation)? Is there a specific gap that the authors would like to fill with this review?

We realize well that this is an extensively covered topic in review articles and we cite many of them. In this review we address latest insights in human HSC culture and expansion in the context of gene therapy approaches. However, in this review, we try to address some essential topics for HSC based gene therapy filed such as:  1) Summary of revised HSCs hierarchy models, 2) Importance of identification of robust markers such as EPCR, ESAM 3) necessity of proper characterization and enrichment of HSCs using more robust markers (eg. CD90, CD49f, CD133, EPCR) rather than just CD34+ . Thus, we here try to give a comprehensive overview of expansion strategies for human HSCs that are targets for gene therapy. This is becoming more relevant with gene repair efforts using nucleases such as CRISPR/Cas9, where HSCs need multiple rounds of division. We therefore think the review is timely and useful.

Therefore the authors believe that this manuscript will provide a valuable source to scientists and healthcare workers in the field

2. The authors have repeated a number of sentences in sections 3, 4, and 5. For example: line 96-98 is the same as line 118-120; line 99-100 repeats in line 131; and line 122-126 repeats in 155-160. The sentences are identical or nearly identical. Given the length of the review, you should avoid repeating sentences like this. Consider combining the sections or elaborating on the original idea without “copy and paste”.

The redundant paragraphs have been removed.

3. Potential typos: line 86 - do you mean “shedding” instead of “shading”? ; line 220 - do you mean “mammals” instead of “mammalian”?; line 242 - do you mean “knock outs” instead of “knocked outs”?; line 257 – do you mean murine “embryo” instead of “embryonic”; line 337 - Fix reference error.

Typos were corrected.

4. Section 9, line 268 – Why do you use Ref 97 and 99 as examples of genetically modified HSC. These studies use small molecules. 

References were updated.

5. On page 2, line 206, you discuss PGE2.  In your discussion and reference to Zon’s work you imply that they used PGE2 in an expansion culture.  The benefit of their strategy is to avoid the need for ex vivo cell expansion. The motivation for a “brief” treatment was that it might not require GMP cell manufacturing facility, which would reduce costs.  Their more recent clinical trial paper is here: Blood 2013 :blood-2013-05-503177; doi: https://doi.org/10.1182/blood-2013-05-503177 . Please modify your discussion to highlight these different strategies.  You should also consider grouping PGE2 differently in your table, as it is not used precisely in same manner as SR-1 for example. This more recent publication also discusses the possible mechanisms by which brief PGE2 treatment of cells might be enhancing transplant outcomes.

We appreciate your suggestion. In lines 309-317 we pointed out the application of PGE2 in shortened or brief expansion of HSC . Also, the effect of PGE2 on improving lentiviral mediated transduction. We also included the reviewer’s useful remarks on benefits of brief ex vivo modulation of stem cells.

6. Regarding Section 10, Concluding remarks: This section is difficult to understand. You may like to consider a different paragraph structure. First paragraph (conclusion) can summarise the unmet needs that would benefit from technologies that enabled in vitro HSC expansion (transplant and gene therapy applications).  Second paragraph – you could move your discussion about the development of better marker profiles to enrich for HSC, deplete specific cell subsets or to identify HSC in expansion products to this paragraph.  Third paragraph – in this paragraph you could consider giving your current optimistic summary of our expanded knowledge of niche biology and expansion strategies and how cumulatively this is likely to help us to identify new pathways forward.

7. With respect to your current concluding remarks, I have questions/comments:

First paragraph – It is not clear what you are proposing regarding the surface markers. Do you propose that the input “HSC population” is wrong or not sufficiently enriched? Or are you saying that we cannot validate the engrafting population after expansion? Is it not useful to expand the progenitor population, which has been shown to provide short-term myeloid support after transplantation? Is it possible that EPCR is specific to UM171 expansion - do you know if this has been validated in other systems? What aspect of this statement refers back to Ref 55? Is there any more recent work?

Here we try to point out the importance of proper enrichment and identification of different subpopulation within input cells for both in vitro and in vivo applications.

We know CD34+ population is heterogenous which contains of LT-HSCs, MPPs and other common progenitors.

Therefore, just enrichment based on CD34+ is not sufficient, and can be misleading.

We believe proper characterization (identification) of input HSCs is important in stem cell based gene therapy, as it provides more accurate information about proportion of different subpopulation.

Indeed expansion of progenitors can be beneficial for gene therapy. Scala et al showed high activity of MPPs during the first wave of reconstitution after translation, however, at the steady state, long term HSCs were taking charge for hematopoietic production compared to MPPs (Scala, et al 2018).

The main issue here is that markers such as CD38, are not useful after culturing of HSCs. Very few markers are, although CD34, Cd90 and CD1333 seem to be constantly expressed. For Cd49f we have not found information.

EPCR was initially identified as a hematopoietic stem cell (HSC) marker by gene expression profiling and RT-PCR analysis of highly purified primitive hematopoietic cells that had been isolated using existing HSC markers in mice (Akashi, et al, 2003, blood)(Balazs et al, 2006. Blood). It may also play a role in HSC self-renewal in mice. These studies suggested that EPCR may be used similarly in human setting as well.

In the recent study by Fares et al, they could purify and sort EPCR (EPCR-/low/+) subset from both uncultured and expanded CD34 +, CD45RA- cord blood cells with UM171. Indeed, UM171 induced the expression of EPCR by 20-56fold (at different time points) on CD34+CD90+CD133+ cells.

Within the same study, they also tested if EPCR is a reliable marker in absence of UM171. The cultured EPCR+ cells in DMSO-supplemented culture, showed long term repopulating activity, suggesting EPCR can be identified in expanded LT-HSCs irrespective of UM171 (Fares et al 2017 blood). We couldn’t find any recent publications on using EPCR in human, while recent paper by (Karimzadeh et al. stem cell tras. 2018) showed EPCR can be used for purification of mouse HSCs.

Second paragraph: 

Line 351 – Consider using the word “identify” instead of “bring up”.

Thanks for the suggestion. It has been implemented.

Round 2

Reviewer 2 Report

The reviewer appreciates the time authors dedicated to the revision, however it is still not clear what are the novel insights this article brings. The 3 points mentioned by the authors to their reply to reviewer #3, are well taken, but also known in the field (especially their last point about identification of more robust HSC markers). Despite the fact that authors aim to "address latest insights in human HSC culture and expansion in the context of gene therapy approaches", a large part of the manuscript refers to murine hematopoiesis (parts 3-5), which leads to confusion. To avoid this issue, authors consider deleting or significantly reducing the text referring to mouse data, and focus only to human hematopoiesis. This will result in a more concise review and will allow authors to better communicate their message to the readers. Towards this direction, it is also recommended to adjust the abstract accordingly. As a result, with all respect to authors' work and efforts, additional work is required in my opinion. 

Minor Points: 

>> Line 367: Spelling error on authors name

>Error has been corrected

It is actually not corrected (line 408 in the updated version), which raises concern about the rest of the references, too.

Author Response

We thank the reviewer for the positive comments. The point of the species differences is well taken. However, many of the advances in HSC expansion are derived from loss of function or gain of function experiments made possible by mouse genetics that until very recently were impossible to perform with human HSCs. Therefore , we have included experiments with mice, but indicated better when referring to mice (by adding text, or indicating in bold fcae when refering to mice). We have adapted the abstract according to the suggestions to indicate the uncertainties still present in expansion of human HSCs.

We apologize for any errors in references. We have digitally reloaded all references from PubMed and inserted them via EndNote into the text. This should avoid mistakes in spelling of names.